# ViViT: Curvature access through the generalized Gauss-Newton's low-rank structure

## Abstract

Curvature in form of the Hessian or its generalized Gauss-Newton (GGN) approximation is valuable for algorithms that rely on a local model for the loss to train, compress, or explain deep networks. Existing methods based on implicit multiplication via automatic differentiation or Kronecker-factored block diagonal approximations do not consider noise in the mini-batch. We present ViViT, a curvature model that leverages the GGN's low-rank structure without further approximations. It allows for efficient computation of eigenvalues, eigenvectors, as well as per-sample first- and second-order directional derivatives. The representation is computed in parallel with gradients in one backward pass and offers a fine-grained cost-accuracy trade-off, which allows it to scale. As examples for ViViT's usefulness, we investigate the directional first- and second-order derivatives during training, and how noise information can be used to improve the stability of second-order methods.

## 1 Introduction & Motivation

The large number of trainable parameters in deep neural networks imposes computational constraints on the information that can be made available to optimization algorithms. Standard machine learning libraries (Abadi et al., 2015; Paszke et al., 2019) mainly provide access to first-order information in the form of *average* mini-batch gradients. This is a limitation that complicates the development of novel methods that may outperform the state-of-the-art: They must use the same objects to remain easy to implement and use, and to rely on the highly optimized code of those libraries. There is evidence that this has led to stagnation in the performance of first-order optimizers (Schmidt et al., 2021). Here, we thus study how to provide efficient access to richer information, namely higher-order derivatives and full statistics of the mini-batch loss.

Recent advances in automatic differentiation (Bradbury et al., 2020; Dangel et al., 2020) have made such information more readily accessible through vectorization of algebraic structure in the differentiated loss. We leverage and extend this functionality to efficiently access curvature in form of the Hessian's generalized Gauss-Newton (GGN) approximation. It offers practical advantages over the Hessian and is established for training (Martens, 2010; Martens & Grosse, 2015), compressing (Singh & Alistarh, 2020), or adding uncertainty to (Ritter et al., 2018b;a; Kristiadi et al., 2020) neural nets. It is also linked theoretically to the natural gradient method (Amari, 2000) via the Fisher information matrix (Martens, 2020, Section 9.2), and has been used to investigate the generalization of neural networks (Jastrzebski et al., 2020; Thomas et al., 2020).

Traditional ways to access curvature fall into two categories. Firstly, repeated automatic differentiation allows for matrix-free exact multiplication with the Hessian (Pearlmutter, 1994) and GGN (Schraudolph, 2002). Iterative linear and eigensolvers can leverage such functionality to compute Newton steps (Martens, 2010; Zhang et al., 2017; Gargiani et al., 2020) and spectral properties (Sagun et al., 2017; 2018; Adams et al., 2018; Ghorbani et al., 2019; Papyan, 2019b; Yao et al., 2019; Granziol et al., 2021) on arbitrary architectures thanks to the generality of automatic differentiation. However, repeated matrix-vector products represent a critical factor for performance.

Secondly, Kronecker-factored approximate curvature (K-FAC) (Martens & Grosse, 2015; Grosse & Martens, 2016; Botev et al., 2017; Martens et al., 2018) constructs an explicit light-weight representation of the GGN based on its algebraic Kronecker structure. The computations are streamlined via gradient backpropagation and the resulting matrices are cheap to store and invert.

This allows K-FAC to scale: It has been used successfully with very large mini-batches (Osawa et al., 2019). One reason for this efficiency is that K-FAC only approximates the GGN's block diagonal, neglecting interactions across layers. Such terms could be useful, however, for other applications, like uncertainty quantification with Laplace approximations (Ritter et al., 2018b;a; Kristiadi et al., 2020) that currently rely on K-FAC. Moreover, due to its specific design for optimization, the Kronecker representation does not become more accurate with more data. It remains a simplification, exact only under assumptions unlikely to be met in practice (Martens & Grosse, 2015). This might be a downside for applications that depend on a precise curvature proxy.

Here, we propose VIVIT (inspired by $VV^\top$ in Equation (3)), a vivid curvature model that leverages the GGN's low-rank structure. Like K-FAC, its representation is computed in parallel with gradients. But it allows a cost-accuracy trade-off, ranging from the *exact* GGN to an approximation that has the cost of a single gradient computation. Our contributions are as follows:

- We highlight the GGN's low-rank structure, and with it a structural limit for the inherent curvature information contained in a mini-batch.
- This low-rank structure allows for efficient computation of various GGN properties: The exact eigenvalue spectrum, including eigenvectors, and per-sample directional derivatives. They enable VIVIT to model curvature noise in a mini-batch, in contrast to existing methods.
- Approximations allow VIVIT to flexibly trade off computational cost and accuracy. We empirically demonstrate scalability on deep neural networks and provide a fully-featured efficient implementation in PYTORCH (Paszke et al., 2019) on top of the BACKPACK (Dangel et al., 2020) package.[1]

Using VIVIT, we illustrate that noise in deep learning poses a challenge for the stability of second-order methods and give a simple example how its quantities can be used to address this problem.

## 2 NOTATION & METHOD

Consider a model $f : \Theta \times \mathbb{X} \to \mathbb{Y}$ and a dataset $\{(\boldsymbol{x}_n, \boldsymbol{y}_n) \in \mathbb{X} \times \mathbb{Y}\}_{n=1}^N$. For simplicity we use $N$ for both the mini-batch and training set size. The network, parameterized by $\boldsymbol{\theta} \in \Theta$, maps a sample $\boldsymbol{x}_n$ to a prediction $\hat{\boldsymbol{y}}_n$. Predictions are scored by a convex loss function $\ell : \mathbb{Y} \times \mathbb{Y} \to \mathbb{R}$ (e.g. cross-entropy or square loss), which compares to the ground truth $\boldsymbol{y}_n$. The training objective $\mathcal{L} : \Theta \to \mathbb{R}$ is the empirical risk

$$\mathcal{L}(\boldsymbol{\theta}) = \tfrac{1}{N} \sum_{n=1}^N \ell(f(\boldsymbol{\theta}, \boldsymbol{x}_n), \boldsymbol{y}_n). \tag{1}$$

We use $\ell_n(\boldsymbol{\theta}) = \ell(f(\boldsymbol{\theta}, \boldsymbol{x}_n), \boldsymbol{y}_n)$ and $f_n(\boldsymbol{\theta}) = f(\boldsymbol{\theta}, \boldsymbol{x}_n)$ for per-sample losses and predictions. For gradients, we write $\boldsymbol{g}_n(\boldsymbol{\theta}) = \nabla_{\boldsymbol{\theta}} \ell_n(\boldsymbol{\theta})$ and $\boldsymbol{g}(\boldsymbol{\theta}) = \nabla_{\boldsymbol{\theta}} \mathcal{L}(\boldsymbol{\theta})$, suppressing $\boldsymbol{\theta}$ if unambiguous. We also set $\Theta = \mathbb{R}^D$ and $\mathbb{Y} = \mathbb{R}^C$ with $D, C$ the model parameter and prediction space dimensions, respectively. For classification, $C$ is the number of classes.

**Hessian & GGN:** Two-fold chain rule application to the split $\ell \circ f$ decomposes the Hessian of Equation (1) into two parts $\nabla_{\boldsymbol{\theta}}^2 \mathcal{L}(\boldsymbol{\theta}) = \boldsymbol{G}(\boldsymbol{\theta}) + \boldsymbol{R}(\boldsymbol{\theta}) \in \mathbb{R}^{D \times D}$; the positive semi-definite GGN

$$\boldsymbol{G} = \tfrac{1}{N} \sum_{n=1}^N (\mathrm{J}_{\boldsymbol{\theta}} f_n)^\top \left(\nabla_{f_n}^2 \ell_n\right) (\mathrm{J}_{\boldsymbol{\theta}} f_n) = \tfrac{1}{N} \sum_{n=1}^N \boldsymbol{G}_n \tag{2}$$

and a residual $\boldsymbol{R} = \frac{1}{N} \sum_{n=1}^N \sum_{c=1}^C \left(\nabla_{\boldsymbol{\theta}}^2 [f_n]_c\right) [\nabla_{f_n} \ell_n]_c$. Here, we use the Jacobian $\mathrm{J}_{\boldsymbol{a}} \boldsymbol{b}$ that contains partial derivatives of $\boldsymbol{b}$ with respect to $\boldsymbol{a}$, $[\mathrm{J}_{\boldsymbol{a}} \boldsymbol{b}]_{ij} = \partial [\boldsymbol{b}]_i / \partial [\boldsymbol{a}]_j$. As the residual may alter the Hessian's definiteness – an undesirable property for many applications – we focus on the GGN.

**Low-rank structure:** By basic inequalities, Equation (2) has $\mathrm{rank}(\boldsymbol{G}) \leq NC$.[2] To make this explicit, we factorize the positive semi-definite Hessian $\nabla_{f_n}^2 \ell_n = \sum_{c=1}^C \boldsymbol{s}_{nc} \boldsymbol{s}_{nc}^\top$, where $\boldsymbol{s}_{nc} \in \mathbb{R}^C$ and denote its backpropagated version by $\boldsymbol{v}_{nc} = [\mathrm{J}_{\boldsymbol{\theta}} f_n]^\top \boldsymbol{s}_{nc} \in \mathbb{R}^D$. Absorbing sums into matrix multiplications, we arrive at the GGN's outer product representation that lies at the heart of VIVIT,

$$\boldsymbol{G} = \tfrac{1}{N} \sum_{n=1}^N \sum_{c=1}^C \boldsymbol{v}_{nc} \boldsymbol{v}_{nc}^\top = \boldsymbol{V} \boldsymbol{V}^\top \text{ with } \boldsymbol{V} = \tfrac{1}{\sqrt{N}} (\boldsymbol{v}_{11} \quad \boldsymbol{v}_{12} \quad \dots \quad \boldsymbol{v}_{NC}) \in \mathbb{R}^{D \times NC}. \tag{3}$$

---

[1] Code available at https://github.com/PwLo3K46/vivit.

[2] We assume the overparameterized deep learning setting ($NC < D$) and suppress the trivial rank bound $D$.

$V$ allows for *exact computations* with the explicit GGN matrix, at linear rather than quadratic memory cost in $D$. We first formulate the extraction of relevant GGN properties from this factorization, before addressing how to further approximate $V$ to reduce memory and computation costs.

## 2.1 COMPUTING THE FULL GGN EIGENSPECTRUM

Each GGN eigenvalue $\lambda \in \mathbb{R}$ satisfies the characteristic polynomial $\det(\boldsymbol{G} - \lambda \boldsymbol{I}_D) = 0$ with identity matrix $\boldsymbol{I}_D \in \mathbb{R}^{D \times D}$. Leveraging the VIVIT factorization of Equation (3) and the matrix determinant lemma, the $D$-dimensional eigenproblem reduces to that of the much smaller Gram matrix $\tilde{\mathbf{G}} = \boldsymbol{V}^\top \boldsymbol{V} \in \mathbb{R}^{NC \times NC}$ which contains pairwise scalar products of $\boldsymbol{v}_{nc}$ (see Appendix A.1),

$$\det(\boldsymbol{G} - \lambda \boldsymbol{I}_D) = 0 \quad \Leftrightarrow \quad (-\lambda)^{D-NC} \det(\tilde{\mathbf{G}} - \lambda \boldsymbol{I}_{NC}) = 0 \,. \tag{4}$$

With at least $D - NC$ trivial solutions that represent vanishing eigenvalues, the GGN curvature is flat along most directions in parameter space. Nontrivial solutions that give rise to curved directions are fully-contained in the Gram matrix, and hence much cheaper to compute. For example, the left panel of Figure 1a visualizes the *full, exact* GGN's empirical spectral density on a mini-batch for a deep convolutional neural net on CIFAR-10. It reproduces the characteristics that have been reported by numerous works, e.g. Sagun et al. (2018): An extensive amount of vanishing or small eigenvalues and a small number of larger outliers.

Despite these various Hessian spectral studies which rely on iterative eigensolvers and implicit matrix multiplication (Sagun et al., 2017; 2018; Adams et al., 2018; Ghorbani et al., 2019; Papyan, 2019b; Yao et al., 2019; Granziol et al., 2021), we are not aware of works that extract the exact GGN spectrum from its Gram matrix. In contrast to those techniques, this matrix can be computed in parallel with gradients in a single backward pass, which results in less sequential overhead. In fact, our approach allows for plots like Figure 1 to be efficiently live-monitored during training, which may be interesting for practitioners that seek to better understand their model (Schneider et al., 2021).

Eigenvalues themselves can help identify reasonable hyperparameters, like learning rates. But we can also reconstruct the associated eigenvectors in parameter space. These are directions along which curvature information is contained in the mini-batch. Let $\tilde{\mathbb{S}}_+ = \{(\lambda_k, \tilde{\mathbf{e}}_k) \,|\, \lambda_k \neq 0, \tilde{\mathbf{G}} \tilde{\mathbf{e}}_k = \lambda_k \tilde{\mathbf{e}}_k\}_{k=1}^K$ denote the nontrivial Gram spectrum with orthonormal eigenvectors $\tilde{\mathbf{e}}_j^\top \tilde{\mathbf{e}}_k = \delta_{jk}$ ($\delta$ represents the Kronecker delta and $K = \operatorname{rank}(\boldsymbol{G})$). Then, the transformed set of vectors $\boldsymbol{e}_k = {}^1\!/\!\sqrt{\lambda_k} \boldsymbol{V} \tilde{\mathbf{e}}_k$ are orthonormal eigenvectors of $\boldsymbol{G}$ associated to eigenvalues $\lambda_k$ (see Appendix A.2),

$$\forall (\lambda_k, \tilde{\mathbf{e}}_k) \in \tilde{\mathbb{S}}_+ : \quad \tilde{\mathbf{G}} \tilde{\mathbf{e}}_k = \lambda_k \tilde{\mathbf{e}}_k \implies \boldsymbol{G} \boldsymbol{V} \tilde{\mathbf{e}}_k = \lambda_k \boldsymbol{V} \tilde{\mathbf{e}}_k \,. \tag{5}$$

The eigenspectrum provides access to the GGN's pseudo-inverse based on $V$ and $\tilde{\mathbb{S}}_+$, required by e.g. second-order methods (see Section 4.2). As we show next, quadratic models defined through the GGN naturally decompose along the eigenvectors from $\mathbb{S}_+ = \{(\lambda_k, \boldsymbol{e}_k) \,|\, \lambda_k \neq 0, \boldsymbol{G} \boldsymbol{e}_k = \lambda_k \boldsymbol{e}_k\}_{k=1}^K$.

## 2.2 COMPUTING DIRECTIONAL DERIVATIVES

Various algorithms update their solution by constructing a local quadratic approximation of the loss landscape. For instance, optimization methods adapt their parameters by stepping to the local proxy's minimum. Such quadratic models are based on a second-order Taylor expansion and hence require some form of curvature. Let $q(\boldsymbol{\theta})$ denote a quadratic model for the loss around position $\boldsymbol{\theta}_t \in \Theta$ that uses curvature represented by the GGN,

$$q(\boldsymbol{\theta}) = \operatorname{const} + (\boldsymbol{\theta} - \boldsymbol{\theta}_t)^\top \boldsymbol{g}(\boldsymbol{\theta}_t) + \frac{1}{2}(\boldsymbol{\theta} - \boldsymbol{\theta}_t)^\top \boldsymbol{G}(\boldsymbol{\theta}_t)(\boldsymbol{\theta} - \boldsymbol{\theta}_t) \,. \tag{6}$$

At its base point $\boldsymbol{\theta}_t$, the shape of $q$ along an arbitrary normalized direction $\boldsymbol{e} \in \Theta$ (i.e. $\|\boldsymbol{e}\| = 1$) is determined by the local gradient and curvature. Specifically, the projection of Equation (6) onto $\boldsymbol{e}$ gives rise to the first-and second-order directional derivatives

$$\gamma_{\boldsymbol{e}} = \boldsymbol{e}^\top \nabla_{\boldsymbol{\theta}} q(\boldsymbol{\theta}_t) = \boldsymbol{e}^\top \boldsymbol{g}(\boldsymbol{\theta}_t) \in \mathbb{R} \,, \tag{7a}$$

$$\lambda_{\boldsymbol{e}} = \boldsymbol{e}^\top \nabla_{\boldsymbol{\theta}}^2 q(\boldsymbol{\theta}_t) \boldsymbol{e} = \boldsymbol{e}^\top \boldsymbol{G}(\boldsymbol{\theta}_t) \boldsymbol{e} \in \mathbb{R} \,. \tag{7b}$$

As $\boldsymbol{G}$'s characteristic directions are its eigenvectors, they form a natural basis for the quadratic model. Denoting $\gamma_k = \gamma_{\boldsymbol{e}_k}$ and $\lambda_k = \lambda_{\boldsymbol{e}_k}$ the directional gradients and curvatures along eigenvector $\boldsymbol{e}_k$, we see from Equation (7b) that the directional curvature indeed coincides with the GGN's eigenvalue.

In addition to the mean gradient and curvature along an eigenvector $\boldsymbol{e}_k$, we can expand the sum over samples from Equation (1) to obtain the per-sample contributions to each derivative. Let $\gamma_{nk}$ and $\lambda_{nk}$ denote these first- and second-order derivatives contributions of sample $\boldsymbol{x}_n$ in direction $k$, i.e.

$$\gamma_{nk} = \boldsymbol{e}_k^\top \boldsymbol{g}_n = \frac{\tilde{\boldsymbol{e}}_k^\top \boldsymbol{V}^\top \boldsymbol{g}_n}{\sqrt{\lambda_k}}, \tag{8a}$$

$$\lambda_{nk} = \boldsymbol{e}_k^\top \boldsymbol{G}_n \boldsymbol{e}_k = \frac{\tilde{\boldsymbol{e}}_k^\top \boldsymbol{V}^\top \boldsymbol{V}_n \boldsymbol{V}_n^\top \boldsymbol{V} \tilde{\boldsymbol{e}}_k}{\lambda_k} = \frac{\|\boldsymbol{V}_n^\top \boldsymbol{V} \tilde{\boldsymbol{e}}_k\|^2}{\lambda_k}, \tag{8b}$$

where $\boldsymbol{V}_n \in \mathbb{R}^{D \times C}$ is the VIVIT factor of $\boldsymbol{G}_n$ corresponding to a scaled sub-matrix of $\boldsymbol{V}$ with fixed sample index. Note that directional derivatives can be evaluated efficiently with the Gram matrix eigenvectors without explicit access to the associated directions in parameter space. In Equation (7) gradient $\boldsymbol{g}$ and curvature $\boldsymbol{G}$ are sums over $\boldsymbol{g}_n$ and $\boldsymbol{G}_n$, respectively. This structure also carries over to the directional derivatives, i.e. $\gamma_k = 1/N \sum_{n=1}^N \gamma_{nk}$ and $\lambda_k = 1/N \sum_{n=1}^N \lambda_{nk}$.

Access to per-sample directional gradients $\gamma_{nk}$ and curvatures $\lambda_{nk}$ along $\boldsymbol{G}$'s natural directions is a distinct feature of VIVIT. Not only do these quantities provide geometric information about the local loss landscape but also about its *directional stochasticity* over the mini-batch. Incorporating such knowledge about the noise into algorithms that rely on quadratic models provides a promising way to increase their performance and stability. In Section 4.2 we show how to use this information to make second-order optimization methods more robust against noise.

## 2.3 COMPUTATIONAL COMPLEXITY

So far, we have formulated the computation of GGN eigenvalues (Equation (4)) including eigenvectors (Equation (5)) and per-sample directional derivatives (Equation (8)). Now, we analyze their computational complexity in more detail to identify critical performance factors. Those limitations can effectively be addressed with approximations that allow the costs to be decreased in a fine-grained fashion. We evaluate their effectiveness on deep neural networks to demonstrate that VIVIT scales.

**Relation to gradient computation:** Machine learning libraries are optimized to backpropagate signals $1/N \nabla_{f_n} \ell_n$ and accumulate the result into the mini-batch gradient $\boldsymbol{g} = 1/N \sum_{n=1}^N [\mathsf{J}_{\boldsymbol{\theta}} f_n]^\top \nabla_{f_n} \ell_n$. Each column $\boldsymbol{v}_{nc}$ of $\boldsymbol{V}$ also involves applying the Jacobian, but to a different vector $\boldsymbol{s}_{nc}$ from the loss Hessian's symmetric factorization. For popular loss functions, like square and cross-entropy loss, this factorization is analytically known and available at negligible overhead. Hence, computing $\boldsymbol{V}$ basically costs $C$ gradient computations as it involves $NC$ backpropagations, while the gradient requires $N$. However, the practical overhead is expected to be smaller: Computations can re-use information from the Jacobians and enjoy significant additional speedup on parallel processors like GPUs. Because our implementation relies on BACKPACK's vectorized Jacobians, we expect similar run time performance as its second-order extensions which have the same backpropagation.

**Stage-wise discarding $\boldsymbol{V}$ (GGN eigenvalues & directional derivatives):** The columns of $\boldsymbol{V}$ correspond to backpropagated vectors. During backpropagation, sub-matrices of $\boldsymbol{V}$, associated to parameters in the current layer, become available once at a time. As many of the above GGN properties can be computed by contraction and accumulation of these stage-wise VIVIT factors, they can be discarded immediately. This allows for memory savings without any approximations.

One example is the Gram matrix $\tilde{\mathbf{G}}$ formed by pairwise scalar products of $\{\boldsymbol{v}_{nc}\}_{n=1,c=1}^{N,C}$ in $\mathcal{O}((NC)^2 D)$ operations. The spectral decomposition $\tilde{\mathbb{S}}_+$ has additional cost of $\mathcal{O}((NC)^3)$. Similarly, the terms for the directional derivatives in Equation (8) can be built up stage-wise: First-order derivatives $\{\gamma_{nk}\}_{n=1,k=1}^{N,K}$ require the vectors $\{\boldsymbol{V}^\top \boldsymbol{g}_n \in \mathbb{R}^{NC}\}_{n=1}^N$ that cost $\mathcal{O}(N^2 CD)$ operations. Second-order derivatives are basically for free, as $\{\boldsymbol{V}_n^\top \boldsymbol{V} \in \mathbb{R}^{C \times NC}\}_{n=1}^N$ is available from $\tilde{\mathbf{G}}$.

**GGN eigenvectors:** Raising one Gram matrix eigenvector $\tilde{\mathbf{e}}_k$ to the GGN eigenvector $\boldsymbol{e}_k$ through application of $\boldsymbol{V}$ (Equation (5)) costs $\mathcal{O}(NCD)$ operations. However, repeated application of $\boldsymbol{V}$ can be avoided for raising weighted sums of the form $\sum_k (c_k/\sqrt{\lambda_k}) \boldsymbol{e}_k$ with arbitrary weights $c_k \in \mathbb{R}$. The summation can be performed in the Gram space at negligible overhead, and only a single vector $\sum_k c_k \tilde{\mathbf{e}}_k$ needs to be transformed. For instance, this allows for efficient aggregation of Newton steps along directions in the Gram space before transforming them to parameter space (see Section 4.2).

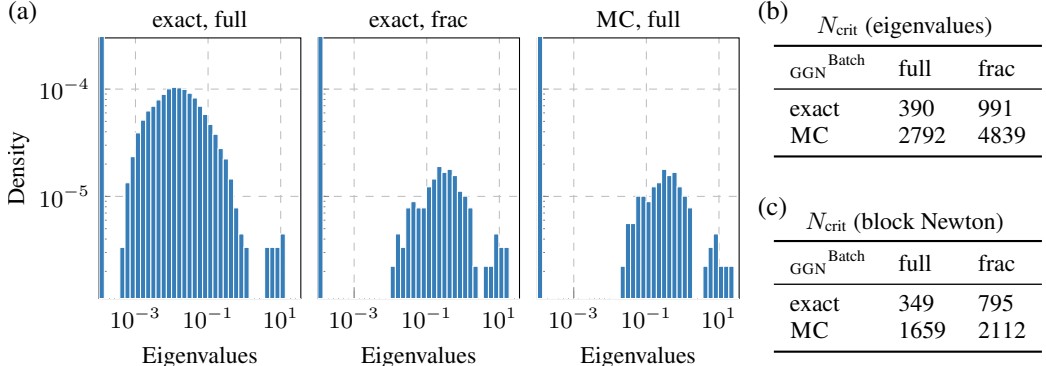

Figure 1: **Reducing costs with curvature sub-sampling, MC-sampling, and parameter groups.**
(a) GGN eigenvalue distribution of the 3C3D architecture on CIFAR-10 ($D = 895{,}210, C = 10$)
for settings with different costs on a mini-batch of size $N = 128$ (Schneider et al., 2019). From left
to right: Exact GGN on the full batch, exact GGN on a batch fraction ($1/8$, as in Zhang et al. (2017)),
MC approximation of the GGN on the full batch. (b) Maximum batch size $N_{\mathrm{crit}}$ on a GeForce RTX
2080 Ti (11 GB) for a standard gradient computation and the GGN spectrum and (c) computing exact
Newton steps with layer-wise parameter groups. More architectures and details in Appendix B.1.

## 2.4 APPROXIMATIONS & IMPLEMENTATION

Although the GGN's representation by $V$ has linear memory cost in $D$, it requires memory equivalent
to $NC$ model copies. Of course, this is infeasible for many networks and data sets, e.g. IMAGENET
($C = 1000$). So far, our formulation was concerned with *exact* computations. We now present
approximations that allow $N$ and $C$ in the above cost analysis to be replaced by smaller numbers,
enabling VIVIT to trade-off accuracy and performance.

**Curvature sub-sampling & MC approximation:** To reduce the scaling in $C$, we can approximate
the factorization $\nabla^2_{f_n} \ell_n(\boldsymbol{\theta}) = \sum_{c=1}^{C} \boldsymbol{s}_{nc} \boldsymbol{s}_{nc}^\top$ by a smaller set of vectors. One principled approach
is to draw MC samples $\{\tilde{\boldsymbol{s}}_{nm}\}$ such that $\mathbb{E}_m[\tilde{\boldsymbol{s}}_{nm}\tilde{\boldsymbol{s}}_{nm}^\top] = \nabla^2_{f_n} \ell_n(\boldsymbol{\theta})$ as in Dangel et al. (2020).
This reduces the scaling of backpropagated vectors from $C$ to the number of MC samples (1 in the
following). Another commonly used independent approximation to reduce the scaling in $N$ is to
compute curvature only on a subset of mini-batch samples (Byrd et al., 2011; Zhang et al., 2017).

**Parameter groups (block-diagonal approximation):** Some applications, e.g. computing Newton
steps, require $V$ to be kept in memory for performing the transformation into the parameter space.
Still, we can reduce costs by using the GGN's diagonal blocks $\{\boldsymbol{G}^{(i)}\}_{i=1}^{L}$ of each layer, rather than
the full matrix $\boldsymbol{G}$. Such blocks are available during backpropagation and can thus be used and
discarded step by step. In addition to the previously described approximations for reducing the costs
in $N$ and $C$, this technique tackles scaling in $D$.

**Concrete example & implementation details:** To assess the quality of the above approximations,
Figure 1a shows the exact GGN eigenvalue spectrum (left) in comparison to its approximation
through curvature sub-sampling (center) and MC (right). Even though the amount of backpropagated
vectors is reduced by a factor of 8 and 10, respectively, the approximated spectra capture essential
features. We also tabularize the critical batch sizes $N_{\mathrm{crit}}$ at which their computations experience
out-of-memory errors in Figure 1b. On a standard GPU, they exceed the traditional batch size used
for training, even for the exact scheme. Approximations further increase the applicable batch size.[3]

---

[3]The critical batch sizes in Figure 1b and c differ strongly for similar reductions of 8 and 10 in the number of
backpropagated vectors. This is because most neural networks have final layers with many parameters. During
initial stages of backpropagation, expanding $V$ for these weights critically affects peak memory, and thus $N_{\mathrm{crit}}$.
This can be improved by leveraging structure in the Jacobian (see Appendix C.1).

As a concrete example for block-diagonal approximations, we group weights and biases layerwise and compute the exact Newton step implied by the GGN's block-diagonal approximation (similar to Zhang et al. (2017)). Figure 1c shows the critical batch size and supports our approach's scalability.

BACKPACK's functionality allows us to efficiently compute individual gradients and $V$ in a single backward pass, using either an exact or MC-factorization of the loss Hessian. To reduce memory consumption, we extend its implementation with a protocol to support mini-batch sub-sampling and parameter groups. By hooks into the package's extensions, we can discard buffers as soon as possible during backpropagation, effectively implementing all discussed approximations and optimizations.

## 3 RELATED WORK

**GGN spectrum & low-rank structure:** Other works point out the GGN's low-rank structure. Botev et al. (2017) present the rank bound and propose an alternative to K-FAC based on backpropagating a decomposition of the loss Hessian. Papyan (2019a) presents the factorization in Equation (3) and studies the eigenvalue spectrum's hierarchy for cross-entropy loss. In this setting, the GGN further decomposes into summands, some of which are then analyzed through similar Gram matrices. These can be obtained as contractions of $\tilde{\mathbf{G}}$, but our approach goes beyond them as it does not neglect terms. We are not aware of works that obtain the exact spectrum *and* leverage a highly-efficient fully-parallel implementation. This may be because, until recently (Bradbury et al., 2020; Dangel et al., 2020), vectorized Jacobians required to perform those operations efficiently were not available.

**Efficient operations with low-rank matrices in deep learning:** Chen et al. (2021) use Equation (3) for element-wise evaluation of the GGN in fully-connected feed-forward neural networks. They also present a variant based on MC sampling. This element-wise evaluation is then used to construct hierarchical matrix approximations of the GGN. VIVIT instead leverages the global low-rank structure that also enjoys efficient eigen-decomposition and linear solves.

Another prominent low-rank matrix in deep learning is the un-centered gradient covariance (sometimes called empirical Fisher). Singh & Alistarh (2020) describe implicit multiplication with its inverse[4] and apply it for neural network compression, assuming the empirical Fisher as Hessian proxy. However, this assumption has limitations, specifically for optimization (Kunstner et al., 2019). In principle though, the low-rank structure also permits the application of our methods from Section 2.

## 4 EXPERIMENTS

Different phases are encountered in the course of neural network training (Frankle et al., 2020). We view those regimes in light of our newly accessible quantities and identify challenges for second-order optimizers. Then, on a simple example, we use those quantities to stabilize such methods.

### 4.1 NOISE DURING TRAINING

The interaction between gradient, curvature, and their stochasticity is crucial for the behavior of optimization methods (Thomas et al., 2020). Here, we aim to identify characteristic features of gradient noise, curvature noise, and their interaction to gain insights into the conditions under which deep learning optimizers are operating. Training different architectures from the DEEPOBS problem set using the baselines from Dangel et al. (2020), we evaluate the GGN's nontrivial eigenvectors and per-sample directional derivatives on a fixed held-out mini-batch with VIVIT. Figure 2 shows results for the 3C3D architecture trained on CIFAR-10 using SGD and cross-entropy loss. A broader evaluation that includes more problems, training with ADAM, and a description of procedural details, is given in Appendix B.2. We make the following observations.

**Directional gradient-curvature correlations:** Figure 2a shows pairs of GGN eigenvalues $\lambda_k$ and the associated directional gradient magnitude $|\gamma_k|$ at different training stages.[5] Like for a quadratic function, gradients and curvatures are positively correlated.

---

[4]For completeness, we describe implicit multiplication with the inverse GGN in Appendix C.2.
[5]We show $|\gamma_k|$ because the directional gradient varies in sign, depending on the eigenvector's orientiation.

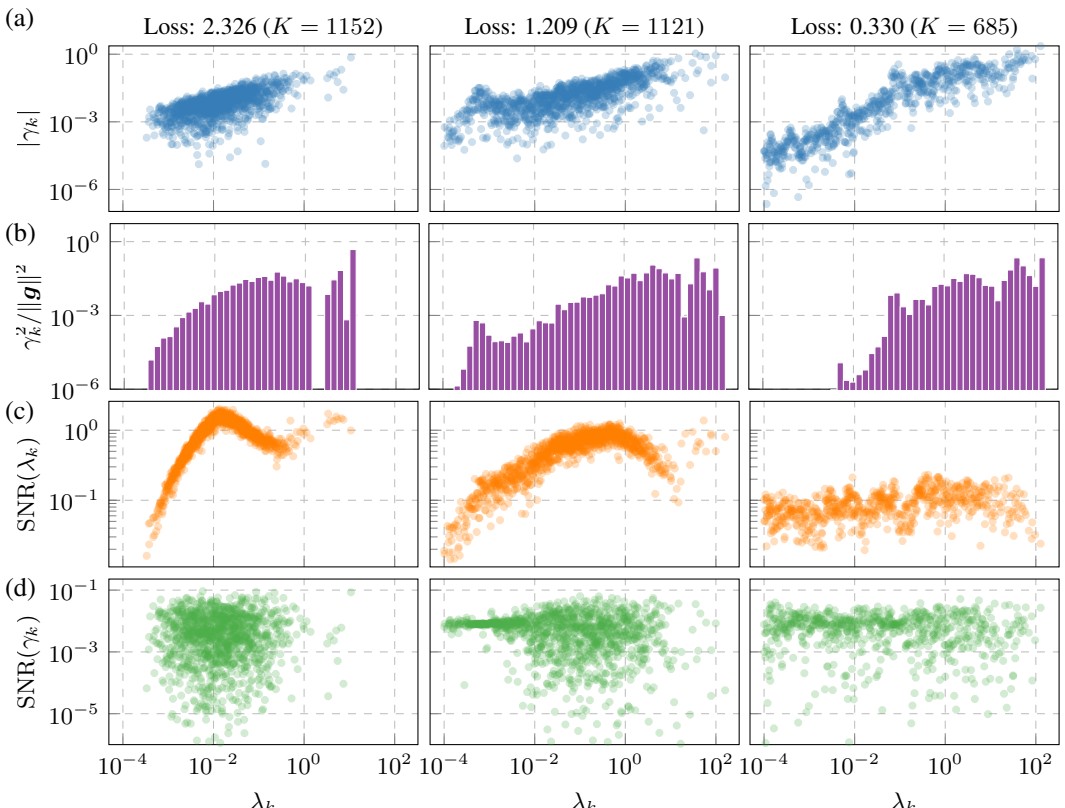

Figure 2: **Gradient, curvature and noise during training.** Columns show the 3C3D architecture at initialization (*left*), an early (epoch 5, *center*), and advanced (epoch 68, *right*) stage of training on CIFAR-10 with SGD (hyperparameters from Schneider et al. (2019); Dangel et al. (2020), details in Appendix B.2). For each direction $k$, characterized by its curvature $\lambda_k$, we monitor (a) the directional gradient magnitude; (b) gradient-eigenvector alignment; (c,d) SNRs of curvatures and gradients.

**High gradient overlap with top eigenspace & shrinking of non-trivial eigenspace:** Similar to Gur-Ari et al. (2018), we observe high alignment between the mini-batch gradient and the GGN's top eigenspace. Specifically, we compute the normalized gradient overlap $\gamma_k^2/\|\boldsymbol{g}\|^2$ with direction $k$. Due to their additivity, gradient overlaps from multiple directions can be grouped by summation. The histograms in Figure 2b show such overlaps: Each bin summarizes directions of similar curvature. The accumulated normalized overlap of the gradient with these directions is shown as the height of the histogram bar. The gradient aligns mostly with high-curvature directions. In particular, its overlap with flat directions, $1 - \sum_{k=1}^{K} \gamma_k^2/\|\boldsymbol{g}\|^2$, is smaller than the axis limits. We observe this alignment throughout training even though the GGN's active sub-space dimension $K$ decreases.

**Curvature signal vanishes during training & gradient signal is consistently small:** To quantify the noise in both the directional gradients and curvatures, we compute their signal-to-noise ratios (SNR). It is given by the squared empirical mean divided by the empirical variance of the mini-batch samples $\{\lambda_{nk}\}_{n=1}^{N}$ and $\{\gamma_{nk}\}_{n=1}^{N}$ for each non-trivial direction $k$.

Figures 2c,d show the evolution of both ratios. At early stages, the curvature signal along some of the high-curvature directions and parts of the bulk dominates over the noise ($\mathrm{SNR}(\lambda_k) > 1$), i.e. curvatures of per-sample GGNs are similar. As training proceeds, the signal decreases until all directions are dominated by noise ($\mathrm{SNR}(\lambda_k) < 1$). In comparison, the directional gradients do not exhibit such a pattern. They are always strongly corrupted by noise ($\mathrm{SNR}(\gamma_k) < 1$), irrespective of the curvature, even though their mean $\gamma_k$ correlates with $\lambda_k$ (Figure 2a).

On the one hand, the high overlap between gradient and the top GGN eigenspace encourages gradient pre-conditioning with the GGN pseudo-inverse for second-order optimization, as negligible gradient

information will be projected out. On the other hand, however, noise in both the gradients and curvatures, which can vary among directions and during training over several orders of magnitude, represents a challenge for such methods. A naive Newton step based on the mean gradient and curvature along each direction may be unstable due to large noise, eliminating all previously made progress. We now turn to those methods and investigate how to improve their stability by equipping them with a noise-aware adaptation strategy to act individually along directions.

## 4.2 USE-CASE: IMPROVING DAMPING IN SECOND-ORDER METHODS

**Spectral decomposition of Newton's method into one-dimensional problems:** Second-order optimizers are based on a local quadratic approximation $q$ (cf. Equation (6)) of the objective function. A common approach to make such methods work in practice is to regularize the curvature matrix with a *damping* term $\delta \boldsymbol{I}_D$, $0 < \delta \in \mathbb{R}$. The resulting quadratic is minimized by the update $\Delta \boldsymbol{\theta} = -(\boldsymbol{G} + \delta \boldsymbol{I}_D)^{-1} \boldsymbol{g}$. In the GGN eigenvector basis from $\mathbb{S}_+$ and the trivial spectrum $\mathbb{S}_0 = \{(\lambda_k, \boldsymbol{e}_k) \mid \lambda_k = 0, \boldsymbol{G}\boldsymbol{e}_k = \boldsymbol{0}\}_{k=K+1}^D$ we observe different updates in the respective sub-spaces,

$$\Delta \boldsymbol{\theta} = -(\boldsymbol{G} + \delta \boldsymbol{I}_D)^{-1} \boldsymbol{g} = -\sum_{k=1}^K \frac{\gamma_k}{\lambda_k + \delta} \boldsymbol{e}_k - \sum_{k=K+1}^D \frac{\gamma_k}{\delta} \boldsymbol{e}_k \,. \tag{9}$$

In the non-trivial eigenspace, $\Delta \boldsymbol{\theta}$ takes damped Newton-type steps, while flat directions are updated with SGD at a learning rate $\delta^{-1}$. Motivated by Section 4.1 and Gur-Ari et al. (2018), we omit the SGD update along flat directions in the following due to their negligible overlap with the gradient.

**Global damping versus directional damping:** Due to mini-batching, the gradient and curvature of the quadratic model are corrupted by sub-sampling noise. Hence, the mini-batch averages $\gamma_k = 1/N \sum_{n=1}^N \gamma_{nk}$ and $\lambda_k = 1/N \sum_{n=1}^N \lambda_{nk}$ in Equation (9) may deviate considerably from the true values on the noise-free training loss. This noise can lead to overly large steps and thus cause the optimization procedure to become unstable. Damping helps to increase stability as it decreases the step size. However, its relative effect on the step length in a direction $\boldsymbol{e}_k$ depends sensitively on the scale of the curvature $\lambda_k$. Also, even at comparable curvature scale, stronger damping may be required in the presence of high uncertainty over $\gamma_k$ and $\lambda_k$. It is thus expected that treating all directions identically yields diminishing returns along certain directions while it is required to keep uncertain ones stable. This suggests a *directional* damping, i.e. an update of the form $\Delta \boldsymbol{\theta} = -\sum_{k=1}^K \frac{\gamma_k}{\lambda_k + \delta_k} \boldsymbol{e}_k$.

**Noise-aware directional damping via bootstrap:** We use the quadratic mini-batch model's loss reduction $\mathcal{R}(\delta_k)$ to assess the step $s_k = -\frac{\gamma_k}{\lambda_k + \delta_k} \in \mathbb{R}$ in direction $\boldsymbol{e}_k$,

$$\mathcal{R}(\delta_k) = q(\boldsymbol{\theta}) - q(\boldsymbol{\theta} + s_k \boldsymbol{e}_k) = -s_k \left( \frac{1}{N} \sum_{n=1}^N \gamma_{nk} \right) - \frac{1}{2} s_k^2 \left( \frac{1}{N} \sum_{n=1}^N \lambda_{nk} \right) \,. \tag{10}$$

From observations on a mini-batch, our goal is to minimize the training loss. Hence, we want to choose a damping such that its corresponding update not only reduces $q$, but *consistently* decreases the loss over all other mini-batch models as well. Ideally, one would compute directional derivatives on additional samples for this purpose. Instead, we use a resampling technique based on the non-parametric bootstrap (Efron, 1979) to simulate samples for $\mathcal{R}(\delta_k)$ on other batches, only using information contained in $\gamma_{nk}, \lambda_{nk}$ (details in Appendix B). For a given damping $\delta_k$, this provides an indicator of what reduction in training loss to expect with the respective update. Taking the 5% percentile of the bootstrap-generated samples, we obtain a *confident* lower bound to $\mathcal{R}(\delta_k)$. We then choose the $\delta_k$ that maximizes this lower bound from candidates on a discrete grid.

**Evaluation on a noisy quadratic:** We consider a quadratic loss function $\mathcal{L}(\boldsymbol{\theta}) = \boldsymbol{\theta}^\top \boldsymbol{G} \boldsymbol{\theta}$ with $\boldsymbol{G}_{ii} = i^2$ for $i \in \{1, ..., D = 20\}$ and initialize $\boldsymbol{\theta}$ at $100 \cdot \boldsymbol{1}_D$. At each step, the optimizer observes unit vectors as directions $\{\boldsymbol{e}_k\}_{k=1}^K$, and noisy directional derivatives $\{\gamma_{nk}\}_{n=1,k=1}^{N,K}, \{\lambda_{nk}\}_{n=1,k=1}^{N,K}$. Specifically, we design the noise to provide unbiased samples with constant variance (see Appendix B for details). We compare our directional damping based on Equation (10) with constant damping and SGD. The results reported in Figure 3 were obtained from multiple runs with different random seeds.

With constant damping, there is a trade-off between large steps that may cause instabilities (small damping) and slower progress due to smaller steps (large damping). Both extremes are observed in our experiment. For $\delta = 10^{-4}$, there are unstable runs, as indicated by the loss mean and the maximum final distance to the minimum. For large constant damping, the behavior becomes more

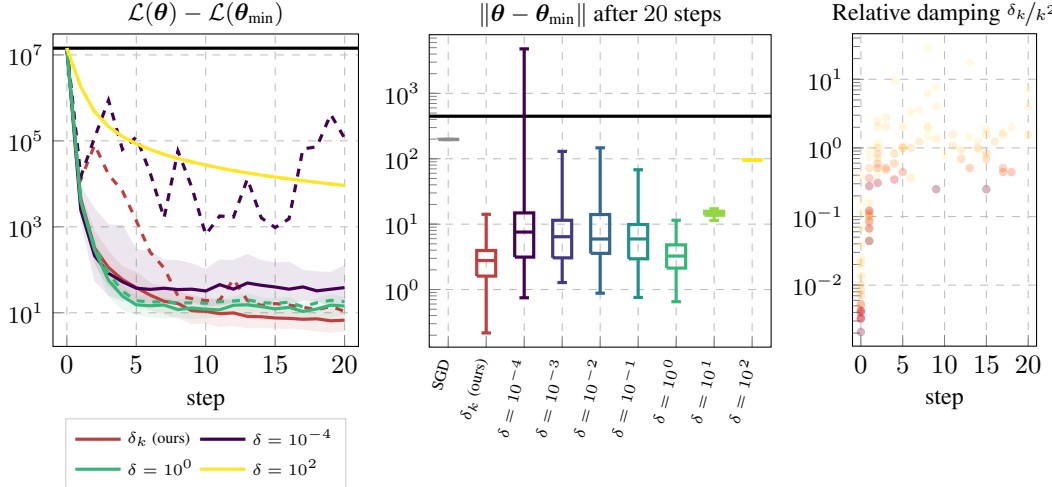

Figure 3: **Comparison of optimizers on noisy quadratic.** Statistics for different optimizers during (*left*) and at the end (*center*) of training over 100 runs. For the loss values, we present mean (dashed line), median (solid line) and a confidence interval (shaded area) between the lower and upper quartile. Boxplot whiskers range from the distribution's minimum to the maximum and boxes indicate lower and upper quartile, enclosing the median. Horizontal black lines indicate the initial loss/distance. *Right*: Relative dampings of our adaptive method. Darker shades indicate larger curvatures.

stable and increasingly resembles SGD. Compared to constant damping, the directional noise-aware damping $\delta_k$ provides the smallest median final distance to the minimum with relatively low variance.

The right panel of Figure 3 shows the directional damping $\delta_k$ in relation to the underlying true curvature $k^2$ for a single run. We make two observations: Firstly, the relative damping increases over time to compensate for the vanishing gradient signal as the optimization approaches the minimum. Secondly, since constant noise is applied, the SNR in low curvature directions (yellow) is smaller than in high-curvature directions (red). This is also reflected in the directional damping, which tends to assign larger dampings to the noisier (yellow) directions. The directional bootstrap damping works as expected and yields stable runs with consistently good final performance.

We provided evidence that ViViT's quantities may be required for the improvement of second-order methods. As the main contribution of this paper is in delivering these quantities, we *deliberately* designed simplistic experiments — a full derivation and empirical evaluation would amount to a separate paper of its own right (and would be incommensurate with the space limitations).

## 5  CONCLUSION

We have presented ViViT, a curvature model based on the low-rank structure of the Hessian's generalized Gauss-Newton approximation. This structure allows for efficient extraction of curvature properties, such as the full eigenvalue spectrum and directional gradients and curvatures along the associated eigenvectors. In contrast to alternatives, ViViT offers statistics of these directional derivatives across the mini-batch, and thus a rich noise model.

We demonstrated the utility of these new quantities by studying noise characteristics of representative deep learning tasks. We find that they pose challenges to the stability of second-order methods, and showed, in a simplistic toy model, how ViViT can provide quantities to improve their stability.

ViViT's representation is efficiently computed in parallel with gradients during a single backward pass. As it mainly relies on vectorized Jacobians, it is even general enough to be integrated into existing machine learning libraries in the future. For the moment, we provide an efficient open-source implementation in PyTorch by extending the existing BACKPACK library.

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
