# OpenReview forum: "ViViT: Curvature access through the generalized Gauss-Newton's low-rank structure"
_ICLR.cc/2022/Conference — ICLR 2022 Submitted_

### Official Review · Reviewer_9mCz · 2021-10-23

**Correctness:** 3
**Technical Novelty And Significance:** 2
**Empirical Novelty And Significance:** 2
**Recommendation:** 5
**Confidence:** 4

**Main Review:**

Strengths:
1. The paper proposes a method tailored to work efficiently on state-of-the-art auto differentiation tools and GPU hardware. Such tools are essential in better understanding various deep learning phenomena, improving optimization algorithms and in aiding practitioners with hyperparameter choices.
2. The observations on the alignment of gradients with top eigendirections correlates well with recent results on deep neural network loss landscapes, and I find the discussion on the evolution of directional gradient and curvature SNR interesting. This direction can be very promising in designing practical second-order optimizers that scale to contemporary models and datasets.
3. The directional damping scheme is an interesting idea, especially the adjustment of damping based on the stochastic uncertainty in a given eigendirection.

Weaknesses:
1. Scalability to practical settings: It is not clear whether the proposed method scales for contemporary architectures and large-scale datasets. Since the low-rank structure of the GGN has been investigated before by others (as the authors also pointed out), the main contribution of this paper is the compute/memory efficiency and scalability of the proposed tool. However, the experiments are performed on CIFAR-10 with a small, 6-layer neural network and it is not clear how it would scale to architectures such as ResNet and datasets such as ImageNet. In case the method does not work well in these scenarios, the applicability of the tool is limited.
2. Overhead: In-depth study on the compute and memory overhead of the method would be valuable. For example statements such as 'However, the practical overhead is expected to be smaller' should be backed by concrete experiments. Even though the memory requirements for different accuracy settings are shown, it would be useful to see how it translates to wall-clock time overhead.
3. Accuracy of the approximation: It is clear that an approximation of V is necessary to make the tool feasible in practical settings. However, the paper doesn't investigate the accuracy of the approximate spectra obtained from the method. Authors claim that the main contribution of the paper is delivering directional derivatives and GGN curvatures, but the accuracy of those quantities is not verified in any way in the paper. Providing some guarantees on the accuracy would be crucial, especially since the method is proposed as an analysis tool. For instance, taking a look at Figure 1 shows that the exact GGN and MC approximate spectra show similar characteristics, but how large is the difference in the top eigenspaces? Small errors in top eigendirections can greatly impact downstream applications where the quantities are used. An in-depth quantitative analysis of the cost savings-accuracy trade-off would be essential.
Furthermore, I am concerned about the accuracy of the method when applied to datasets with larger number of classes. For instance, the CIFAR-100 results in Figure S.5 show that the MC approximate spectrum looks very different from the exact spectrum, even qualitatively. This can be a serious issue when scaling the method to datasets such as ImageNet.
4. Damping: The directional damping method is interesting and is a neat application for the proposed tool, but 1) if it is meant only as a proof-of-concept, the discussion takes up a very significant portion of the paper (almost 2 pages) that could be used for supporting the main claims of the paper or 2) if it is meant as a main contribution and a practical method, then much more in-depth studies are needed to establish whether it improves damping in second-order methods.

Minor: What does K denote exactly (first appearing in 4.1)? Is K=NC?

**Summary Of The Paper:**

This paper highlights how the low-rank structure of the generalized Gauss-Newton (GGN) approximation of the Hessian can be used as a computationally efficient tool to study the loss landscape of deep neural networks. In particular, authors discuss methods to compute the full spectrum of the GGN and thus providing access to per-sample directional gradients and curvature approximation. Through the lens of the GGN spectrum, authors make observations on the geometry of the loss landscape and its evolution during training, and propose an adaptive damping technique for second-order optimizers that utilizes the GGN curvature information.

**Summary Of The Review:**

The paper relies heavily on different forms of approximations to make the technique feasible, however  discussion on the accuracy of the approximated quantities is lacking, therefore it is difficult to judge how useful the obtained quantities are. Moreover, the scalability of the method to contemporary, practical architectures and datasets is not well-supported, thus the significance of the results is limited. Therefore, I recommend rejection of the paper in its current form. However, I believe that the paper has merit and some interesting contributions, and I am open to increasing my score if the authors address my concerns.

---

### Official Review · Reviewer_VGMZ · 2021-11-02

**Correctness:** 3
**Technical Novelty And Significance:** 2
**Empirical Novelty And Significance:** 3
**Recommendation:** 5
**Confidence:** 2

**Main Review:**

Strengths:

By leveraging the GGN’s low-rank structure, one can efficiently computate the eigenvalues, eigenvectors,  per-sample first- and second-order directional derivatives, and features.

Weakness:
1) The technical novelty of using low-rankness of GGN's seems a bit limited. It is suggested to highlight the difference as well as technical difficulty with existing works in exploiting the low-rank structure for fast computation of quantities like eigenvalues.
2) Give more clear theoretical explanations of why the residual in Eq. (2)  is simply ignored.
3) The theoretical analysis of why noise information improves the stability of second-order methods seems insufficient.

**Summary Of The Paper:**

This paper leverages the GGN’s low-rank structure, which allows for efficient computation of eigenvalues, eigenvectors,  per-sample first- and second-order directional derivatives, and parallel feature computation. The empirical performance is encouraging.

**Summary Of The Review:**

This paper proposes to use the GGN’s low-rank structure for fast and sound model learning. The empirical performance in promising. However, the technical novelty of using low-rankness of GGN's seems limited and the technical significance seems not sufficiently strong.

---

### Official Review · Reviewer_GBwe · 2021-11-02

**Correctness:** 3
**Technical Novelty And Significance:** 3
**Empirical Novelty And Significance:** 3
**Recommendation:** 6
**Confidence:** 4

**Main Review:**

Strengths:
1.	It is novel that the proposed curvature model VIVIT extracts full spectrum from the exact GGN matrix (on each mini-batch) and provides per-sample directional derivatives and curvatures, which may be of interest to future work on their own.
2.	The proposed usage of VIVIT in second-order methods, i.e., the directional damping, is novel and deals with the difficulty of step size selection.
3.	Overall, the paper is well organized and clearly written. Thorough discussions on computational complexity and implementation details make this work practical.

Weaknesses:
1.	Why is the GGN matrix a good approximation of the true Hessian matrix? Does the GGN matrix well capture the top eigenvalues of the true Hessian? What is left in the residual matrix $R$? It would be better to have a discussion of the quality of such GGN approximation or demonstrate it in some experiment.
2.	It would be better to compare the obtained eigenvalue spectrum (and eigen-vectors) with some of the existing curvature approximation methods in the experiment. Though these methods may not provide per-sample curvature estimation, it is still necessary to compare with them to show the quality and efficiency of VIVIT.
3.	In Figure 3, it would be better to demonstrate how well the proposed directional damping balances convergence speed and stability. Currently in the left plot its performance seems to be not as good as the simple choice $\delta=1$.
4.	It would be better to distinguish between the mini-bath size and the training set size and formally state that VIVIT is for the underdetermined cases with $D > NC$.


**Summary Of The Paper:**

This work proposes a curvature model VIVIT based on generalized Gauss-Newton (GGN) approximation for the training of neural networks with a convex loss function. The low-rank structure of VIVIT allows for efficient eigen-value decomposition, which also gives per-sample directional derivatives and curvatures. To further improve the efficiency, sampling within the mini-batch and among the coordinates of the prediction can be applied to trade off computational cost with accuracy. As an application example, VIVIT is used to provide noise-aware directional damping which improves the stability of second-order methods.

**Summary Of The Review:**

The proposed curvature approximation method is novel and practical and appears to have the potential of being useful to future work. Though there are a few things to be improved, in general I find this work well written, and my current recommendation is marginally above the acceptance threshold.

---

### Official Review · Reviewer_1dgm · 2021-11-05

**Correctness:** 4
**Technical Novelty And Significance:** 1
**Empirical Novelty And Significance:** 3
**Recommendation:** 5
**Confidence:** 4

**Main Review:**

The paper is clearly written and easy to follow.

**Main idea: low-rank structure of the GGN**: The main idea of taking advantage of the low-rank structure of the GGN is clearly motivated as well as the linear algebra manipulations (though somewhat basic) used to get the spectrum of the GGN. I would however not deem it as new as it is e.g. discussed in [1,2], but arguably it is interesting to emphasize how to use this low-rank structure in deep learning algorithms.

**Implementation** It is difficult to judge given that no code is provided (or linked) but it would certainly be very useful to have a good implementation for such quantities. I would be interested in seeing timings (or at least orders of magnitudes) for your experiments on this reasonably large model of section 4.1. I am not sure however that ICLR is the most appropriate conference for implementation papers.

**Per-sample variance analysis** (section 4.1) is new as far as I can tell. It provides very interesting insights about the SNR of the gradients and curvature projected onto the eigenvectors of the GGN. I would have appreciated some more insights with more standard networks (e.g. ResNets)

**Proposed adaptive damping scheme** The information about mini-batch variance is nicely exploited in the proposed adaptive damping scheme for second-order methods. As you rightfully notice, fully analyzing and benchmarking such an algorithm on an actual neural network would require a separate paper, which I would be looking forward to reading. A difficult problem that I can foresee is that even by estimating the GGN on a minibatch, the partial left singular vectors of $\mathbf V$ are of dimension $D$ which can already be quite large.

[1] https://arxiv.org/abs/1905.11675

[2] https://arxiv.org/abs/1806.02958

**Summary Of The Paper:**

This paper describes a way of leveraging the low-rank structure of the generalized Gauss-Newton (GGN) matrix, and a library that extends BackPACK's features in order to efficiently compute:
 - the spectrum (eigenvalues, eigenvectors) of the GGN
 - per sample directional derivatives and curvature of the GGN

They then demonstrate the library on a 900K parameters x 10 classes deep network on CIFAR-10.
They also propose a damping scheme for second-order that only have access to noisy estimates of the curvature.

**Summary Of The Review:**

If it were a purely theory paper, I am not sure that the idea of leveraging the low-rank structure of the GGN makes for a novel enough contribution by itself to be published at ICLR.

It if were a pure implementation paper, I think ICLR would not be the most appropriate venue.

Hence my low rating, for a paper that is otherwise very pleasant to read.

I think that the proposed adaptive damping scheme can however make for a nice follow-up.

---

### Decision · Program_Chairs · 2022-01-20

**Decision:**

Reject

**Comment:**

A method for efficient exact computation of the generalized Gauss-Newton matrix is given. Using this method the authors provide several empirical observations of first and second order statistics of neural networks during training. Additionally the authors use to tool to propose a new damping technique that some reviewers found particularly interesting. Reviewers noted that the low rank decomposition the authors provide is not new, and has been used in prior work, although the trick may not be widely known within the deep learning community. As such novelty is not a strength of the work, and reviewers suggested the authors could strengthen the work with a convincing demonstration that the method can be made to work at scale, as well as providing more detailed run time and memory comparisons with other approaches to calculating the GGN matrix. Although the authors agreed with reviewer suggestions, the paper was not updated during the rebuttal period. As such I recommend the authors resubmit with the proposed revisions.